# Genome-Wide Association Screening Determines Peripheral Players in Male Fertility Maintenance

**DOI:** 10.3390/ijms24010524

**Published:** 2022-12-28

**Authors:** Thomas Greither, Hermann M. Behre, Holger Herlyn

**Affiliations:** 1Center for Reproductive Medicine and Andrology, Martin Luther University Halle-Wittenberg, 06120 Halle, Germany; 2Anthropology, Institute of Organismic and Molecular Evolution (iomE), Johannes Gutenberg University Mainz, 55099 Mainz, Germany

**Keywords:** male infertility, disorder, polymorphism, genome-wide association study, spermatogenesis, spermiogenesis

## Abstract

Deciphering the functional relationships of genes resulting from genome-wide screens for polymorphisms that are associated with phenotypic variations can be challenging. However, given the common association with certain phenotypes, a functional link should exist. We have tested this prediction in newly sequenced exomes of altogether 100 men representing different states of fertility. Fertile subjects presented with normal semen parameters and had naturally fathered offspring. In contrast, infertile probands were involuntarily childless and had reduced sperm quantity and quality. Genome-wide association study (GWAS) linked twelve non-synonymous single-nucleotide polymorphisms (SNPs) to fertility variation between both cohorts. The SNPs localized to nine genes for which previous evidence is in line with a role in male fertility maintenance: *ANAPC1*, *CES1*, *FAM131C*, *HLA-DRB1*, *KMT2C*, *NOMO1*, *SAA1*, *SRGAP2*, and *SUSD2*. Most of the SNPs residing in these genes imply amino acid exchanges that should only moderately affect protein functionality. In addition, proteins encoded by genes from present GWAS occupied peripheral positions in a protein–protein interaction network, the backbone of which consisted of genes listed in the Online Mendelian Inheritance in Man (OMIM) database for their implication in male infertility. Suggestive of an indirect impact on male fertility, the genes focused were indeed linked to each other, albeit mediated by other interactants. Thus, the chances of identifying a central player in male infertility by GWAS could be limited in general. Furthermore, the SNPs determined and the genes containing these might prove to have potential as biomarkers in the diagnosis of male fertility.

## 1. Introduction

An estimated 8–15% of couples worldwide are unintentionally childless, whereby male infertility is assumedly involved in 40–50% of the cases [1,2,3]. Reasons for male infertility are manifold and include deviant transcript and protein abundances in the testis and sperm [4,5,6]. Chromosomal aberrations, such as Klinefelter syndrome (47, XXY), are also well-recognized to negatively affect male fertility [7]. Changes in chromosome structure can elicit infertility too. Prominent examples are AZF (azoospermia factor) regions on the Y chromosome, deletions in which cause severe oligozoospermia and azoospermia [8,9,10]. In fact, 2–10% of infertile men and up to 17% of infertile men with reduced sperm count (<1 mio/mL) are assumed to carry such deletions [11,12,13]. Yet, subtle variation can also affect male fertility. This encompasses single-nucleotide polymorphisms (SNPs), which were initially analyzed in gene-centric approaches [8,14]. Notwithstanding the importance of these studies, one of the major determinants of male fertility, spermiogenesis, is a complex process that relies on the interaction of hundreds to thousands of genes [15]. In fact, the maturation of germ cells requires the interaction of spermiogenesis stages with other cell types, such as Sertoli, Leydig, and peritubular myoid cells. This complexity implicates many possible disturbances [4]. It is therefore reasonable to consider the entire range of possible factors when investigating the causes of impaired spermiogenesis [8,14]. The widespread application of genome-wide screening technologies to the study of diseases and disorders [3] demonstrates that corresponding tools are well-established.

Based on microarrays, it has been shown that copy number variation, especially on the Y chromosome, can lead to spermatogenic failure [16,17,18,19]. Genome-wide screenings also underscored the significance of deletions for impaired spermatogenesis [20,21,22] and sperm malformation [23,24]. Furthermore, several single-nucleotide variants (SNVs) were found to relate to testicular dysgenesis [25]. However, array technology linked a conspicuously large number of SNPs to oligozoospermia and non-obstructive azoospermia [4,22,26,27,28,29]. More recently, array-based genotyping in a Greek population discovered an association of SNPs in gene regulatory RNAs with abnormal sperm count (oligozoospermia), motility (asthenozoospermia), and morphology (teratozoospermia) [30]. Another genome-wide association study (GWAS) revealed links of SNPs in North American Hutterite men with the number of children fathered [31].

Due to the technical progress of the last years, SNPs associated with male infertility are being increasingly determined in next-generation sequencing data. Thus, whole exome sequencing (WES) uncovered associations of SNPs in *BRDT*, *SUN5*, and *PMFBP1* in sperm malformation [32,33,34,35]. Additional WES-based studies associated mutations with non-obstructive azoospermia [11,36,37,38,39]. Despite this development, we are far from having fully comprehended the molecular causes of male infertility. Therefore, unexplained, or idiopathic, infertility remains a frequent diagnosis. Complicating matters, candidate markers may not be universally predictive, as illustrated by the SNPs mentioned above that were associated with childbearing in Hutterite men. Of four re-analyzed SNPs, only some were associated with infertility in Japanese men [40], and none with sperm parameters [41]. This multi-layered picture illustrates that more efforts are needed to elucidate the deeper causes of male infertility in different populations.

Another challenge is to decipher the functional interrelationships of SNP-bearing genes as emerging from GWAS. The relevant genes should interact [1,2], but it can be difficult to clarify the nature of the interaction. Related to this, it remains to be answered if genes harboring associated SNPs are of central or marginal importance for a particular phenotype. In respect to fertility maintenance, a moderate impact appears to be more likely since variants causing infertility cannot be passed on to the next generation naturally [8]. In network reconstruction, corresponding genes or the proteins encoded should primarily occupy peripheral positions, suggestive of reduced functional relevance [2,42]. We have tested these predictions against the background of fertility variation in men of central European origin. For this purpose, we conducted a GWAS on the exomes of, altogether, 100 men. Spermiogram parameters were in the normal range in the fertile group while infertile subjects had reduced sperm quantity and quality. Fertile subjects additionally had already conceived offspring naturally, whereas infertile men were unintentionally childless. Genes containing associated variants were examined in various databases for previous evidence of fertility relevance in men. We additionally investigated whether the corresponding proteins integrate into a larger protein–protein interaction (PPI) network of male fertility maintenance. Especially, we examined whether GWAS might preferentially determine SNPs in genes encoding proteins with peripheral positions in network reconstruction.

## 2. Results

Samples of blood were taken from 100 infertile and normal-fertile men (Figure 1). Infertile probands were involuntarily childless and had been diagnosed with azoospermia, oligozoospermia, oligoasthenozoospermia, and oligoasthenoteratozoospermia. Men in the control cohort were fathers presenting with normozoospermia. Accordingly, parameters relating to sperm amount, motility, and morphology considerably differed between both cohorts. On the contrary, semen pH was almost identical. Furthermore, body-mass index (BMI) was somewhat elevated in infertile probands relative to normal-fertile subjects. Moreover, infertile men were older than fertile ones (Table 1). Considering the corresponding ratios, an overall decreased number of fully functional sperm seemed to be the major theme behind the fertility differences observed, rather than differences in age, BMI, or semen pH.

Present WES yielded a minimum of 6 G of high-quality (Q30 ≥ 80%) 150 bp paired-end (PE) reads for each of the DNA samples provided by, altogether, 100 fertile and infertile men (Appendix A). In total, 840 SNPs showed different allele frequencies between the cohorts. The focus on SNPs genotyped in all 100 men increased the probability of gaining identical allele frequencies and test results (Fisher’s exact) across different loci. This factor probably contributed to the partial stratification of false discovery rates (FDRs) undershooting the 5% significance threshold (Figure 2). Yet, for the SNPs exceeding the threshold there were no signs of stratification. Except for one, these were within the 95% confidence interval under the assumption of a normal distribution, indicative of the independence of the test results (Figure 3).

Downstream, we only kept significantly associated SNPs for which we manually confirmed SNVs, chromosome, position, and gene name in the dbSNP database at the National Center for Biotechnology Information (NCBI) (Figure 1; for more details, see Section 4). For consideration, SNPs additionally had to reside in genes for which expression in male reproductive tissues was demonstrated before. This filtering converged in 12 SNPs with significantly differing allele frequencies between normal-fertile and infertile men (FDR < 0.05, each). In half of these, the otherwise minor allele represented the majority (Table 2; Appendix A). One SNP was intronic and might affect the splicing of *HLA-DRB1* transcript. Correspondingly, the effect of the alternative SNV has the potential to be high according to the Variant Effect Predictor tool at the Ensembl database (Ensembl VEP). The remaining SNPs implicated amino acid exchanges, which Ensembl VEP predicted to moderately affect protein functionality (Table 2). Still, the scoring by the Sorting Intolerant from Tolerant (SIFT) algorithm underlined the ‘deleterious’ nature of the alternative alleles. Likewise, ‘probably damaging’ predominated in effect predictions according to Polymorphism Phenotyping (PolyPhen), followed by ‘benign’ and ‘possibly damaging’ (Appendix A).

The SNPs pinpointed resided in nine genes whereby three genes harbored two SNPs, each (Table 2): *ANAPC1* (anaphase promoting complex subunit 1), *CES1* (carboxylesterase 1), FAM131C (family with sequence similarity 131 member C), *HLA-DRB1* (major histocompatibility complex, class II, DR beta 1), *KMT2C* (lysine methyltransferase 2C), *NOMO1* (NODAL modulator 1), *SAA1* (serum amyloid A1), *SRGAP2* (SLIT-ROBO Rho GTPase activating protein 2), and *SUSD2* (sushi domain containing 2). As mentioned above, all nine genes were known to be expressed in several male reproductive tissues (Table 3). Thus, protein data of five of the genes were detected in up to four male reproductive tissues before, according to The Human Protein Atlas. Yet, lack of corresponding entries in the other four genes was presumably due to the insufficient data. In support of this view, transcripts of all nine genes had been determined for either the testis, prostate, and seminal vesicle (*NOMO1*), or the same tissues and epididymis (all others). Accordingly, the Male Infertility Knowledgebase lists all nine genes under focus to be associated with male fertility disorders, ranging from cryptorchidism via lowered sperm count and quality to immunoinfertility. Thorough evaluation of the underlying data revealed that these assignments were based on deviant methylation patterns, transcript abundances, and allele frequencies in (testis and sperm of) reduced-fertile men (Table 3).

Subsequently, we addressed if the genes emerging from present GWAS might play a central or peripheral role in the development of male fertility impairment. For this purpose, we subjected the corresponding genes to network analysis. STRING v11.5 did not find any PPI between the proteins encoded. Yet, the picture changed when network reconstruction additionally considered genes listed in OMIM (Online Mendelian Inheritance in Man database at NCBI), for their involvement in male infertility, asthenozoospermia, teratozoospermia, and spermatogenic failure. Thus, most of the nine genes in focus were linked through proteins encoded by OMIM-listed genes in the larger network. Notably, none of the proteins encoded by genes from present GWAS occupied a central or intermediary position. Instead, the proteins of seven genes (*ANAPC1*, *CES1*, *HLA-DRB1*, *KMT2C*, *SAA1*, *SRGAP2*, *SUSD2*) localized to the margin of the largest connected component or LCC (Figure 4; Appendix A). Their peripheral localization was reflected in an almost three-fold lower average node degree (1.286) than across all nodes in the LCC (3.682). Thus, present GWAS preferentially determined genes at the margin of the male infertility network.

## 3. Discussion

The present pipeline (Figure 1) resulted in 12 SNPs with significantly differing allele frequencies between fertile and infertile probands (Table 1 and Table 2). Previous evidence is in line with their relevance for male fertility. Indeed, the nine genes harboring the SNPs are expressed in male reproductive tissues (Table 3). Fertility relevance is further suggested by corresponding entries in the Male Infertility Knowledgebase [51] reporting aberrant CpG and methylation patterns and deviant transcript abundances in (testis and sperm of) reduced-fertile men [44,45,48,49,50]. In one of the respective genes (*HLA-DRB1*), there were also previous hints for shifted allele frequencies in men with fertility impairment [46,47]. In fact, human leukocyte antigen (HLA) regions generally appear to contain susceptibility loci for non-obstructive azoospermia as reflected in the results of a GWAS carried out on Chinese men [26]. On the other hand, the genes harboring associated SNPs are unlikely to affect fertility levels directly. Thus, none of the corresponding proteins took a central or intermediary position in the PPI network the backbone of which consisted of proteins or genes with OMIM entries relating to a role in male fertility impairment (Figure 4). Rather, proteins to seven genes emerging from the present investigation (*ANAPC1*, *CES1*, *HLA-DRB1*, *KMT2C*, *SAA1*, *SRGAP2*, *SUSD2*) occupied peripheral positions in the LCC of that network, thus being connected amongst each other through other proteins. An edging position is also likely for the two genes identified herein (*FAM131C*, *NOMO1*), the proteins of which remained unconnected in network reconstruction. We consider it improbable that these will take central positions in a yet-to-be-defined larger network of male fertility disorders. Yet, positional peripherality in a network usually indicates decreased functional importance of the coding gene and vice versa [2,42]. This rule was confirmed for male reproductive genes [52,53], so the genes identified here will probably be peripheral modulators of male fertility, instead of central players in male fertility maintenance.

The functional data available accord with an indirect involvement of the nine genes under investigation in male fertility. Thus, the protein of the zebrafish orthologue to human *NOMO1* is part of a complex that interferes with NODAL signaling and mesodermal patterning [54]. Yet, NODAL signaling has been shown to steer germ cell differentiation and the establishment of seminiferous cords during early testicular development in human embryogenesis [55]. Indeed, NODAL has been found to be secreted by male germ cells and controls Sertoli cell proliferation and function [56]. Notably, the level of the signaling factor, NODAL, raises when another gene that we pinpointed, *KMT2C* (syn. *MLL3*), is lacking [57]. The encoded methyltransferase is supposedly important for early embryogenesis and spermatogenesis [58] by setting methylation marks in nucleosomes [59,60,61]. Consistently, male mice that lack a fully functional enzyme showed reduced fertility [62].

The remainder genes arising from present GWAS could be involved in spermiogenesis according to published data too. For example, testicular transcript abundance of the murine orthologue to *FAM131C* was found to be decreased in male mice displaying impaired spermatogenesis and reduced fertility [63]. The proteins of two additional genes might participate in spermatogenesis via an implication in cell-cycle control, namely SUSD2 in the arrest of G1/S phase transition [64] and ANAPC1 in metaphase/anaphase progression [65]. In accordance with such fundamental functions, there is ample evidence for a role of *SUSD2* in the etiology of infertility, although restricted to females to date (e.g., [66]). But *ANAPC1* is a likely risk locus in the development of Rothmund–Thomson Syndrome which usually includes male and female fertility impairment [67,68]. The gene *CES1* also appears to be involved in spermiogenesis. Indeed, *CES1* was found differentially expressed in cells overexpressing *RHOX* cluster genes which are known to be important for fertility maintenance in men [69,70]. Correspondingly, *Ces1* exhibited altered expression in an infertile murine knock-out strain [71]. In addition, *SRGAP2* might be implicated in spermatogenetic failure [72,73,74], although the gene is primarily known for its role in neuronal migration (Table 2). However, the basic function of the encoded protein is seen in the stimulation of a GTPase (RAC1) steering progressive swimming and competitiveness of spermatozoa in mammals [75].

Not least, evidence for associated SNPs in *SAA1* and *HLA-DRB1* might relate to the protection of spermatogenesis stages from the own immune system [76,77]. In the case of *SAA1*, such a correlation emerges from a case study according to which the mRNAs of proinflammatory genes, such as *SAA1*, were up-regulated in the testicular tubule microenvironment of an infertile man presenting with Fabry’s disease [78]. For *HLA-DRB1*, immunoinfertility belongs to the associated phenotypes listed in the Male Infertility Knowledgebase (Table 3). According to the same database, many of the genes discussed here in the context of spermiogenesis are involved in the development of cryptorchidism. This is not a contradiction since unimpaired spermiogenesis requires lowered temperature as achieved by testis descent [79]. Furthermore, there is evidence that cryptorchidism may facilitate the emergence of antisperm antibodies and thus the development of autoimmunoinfertility [80,81], as we address here for *SAA1* and *HLA-DRB1*. These considerations illustrate that the genes under study could be indirectly involved in male fertility maintenance

## 4. Materials and Methods

### 4.1. Diagnosis and Cohorts

The Ethics Committee of the Medical Faculty at the Martin Luther University Halle (Saale) gave permission for sample collection (approval number 218/14.04.10/2.; decision date: 19.04.2010; date of approved amendment: 25.07.2022). All participants provided written informed consent. Spermiogram parameters were recorded according to the sixth edition of the WHO laboratory manual for the examination and processing of human semen [82]. For simplicity, we use descriptive terms (azoospermia, oligozoospermia, asthenozoospermia, teratozoospermia, and mixed forms) in the present study as established in previous editions of the World Health Organization (WHO) manual. In the infertility cohort, we grouped together men presenting with different forms of fertility impairment. By doing so, we aimed at identifying associated SNPs and genes harboring these, which should have predictive potential for a broad range of infertility forms. Altogether, 100 probands were recruited from the fertility outpatient care of the University hospital Halle (Saale) in Germany. Infertile men (N = 70) presented with azoospermia (N = 5), oligozoospermia (N = 25), asthenozoospermia (N = 8), oligoasthenozoospermia (N = 10), and oligoasthenoteratozoospermia (N = 22). They were unintentionally childless, i.e., their partners did not conceive despite regular unprotected intercourse within at least twelve months. Men in the control cohort (N = 30) presented with normozoospermia, and already had fathered offspring without medical assistance. Both groups did not include patients with AZF deletions or chromosomal translocations. There were no family relatives amongst the probands, which all shared a central European origin (Appendix A).

### 4.2. Lab Work: Sample Processing and Sequencing

Blood samples were supplemented with EDTA (Applichem, Darmstadt, Germany) and ultra-frozen until processing. Upon gentle thawing on ice, genomic DNAs were extracted with DNeasy Blood & Tissue kit (Qiagen, Hilden, Germany) according to the manufacturer’s instruction. Whole exome sequencing (WES) of the libraries (Agilent SureSelect Human All Exon V6; Agilent Technologies, Santa Clara, CA, USA) was conducted on an Illumina platform (NovaSeq 6000, PE150; Illumina, San Diego, CA, USA). A minimum of 6 G raw data were collected per run. The datasets have been deposited at NCBI under BioProject ID PRJNA898129.

### 4.3. SNPs and GWAS

Reads were aligned to the human reference genome GRCh37 (hg19). Subsequently, we discarded synonymous SNVs while keeping missense and splicing ones. Furthermore, we only considered SNPs with minor allele frequency (MAF) <1%, each, in ALL populations from phase III of the 1000 Genomes Project (hg19-1000g2015_all). By doing so, we aimed at filtering for rare SNVs which were more likely to be detrimental. Another requirement was that genotyping had been performed for all 100 individuals. The corresponding pipeline used in-house scripts. Allele frequencies between fertile and infertile probands were compared using Fisher’s exact test in the stats package of R [83]. Resulting p-values were transformed into false discovery rates (FDRs) following Benjamini and Hochberg (1996) [43]. For illustrating FDR levels across the genome, we used the Manhattan plot and Q–Q plot tools at https://www.bioinformatics.com.cn/srplot (accessed on 20 December 2022).

### 4.4. Validation and Effect Prediction

We manually validated annotations by ANNOVAR [84] in NCBI dbSNP [85]. To be kept, a SNP had to be verified in respect to the SNVs involved, chromosome, nucleotide position, and gene name in the dbSNP database at www.ncbi.nlm.nih.gov. We additionally considered expression patterns in male reproductive tissues (testis, prostate, epididymis, seminal vesicle) according to The Human Protein Atlas at www.proteinatlas.org [86]. Genes harboring associated SNPs were also checked for previous evidence of fertility relevance. Especially, we searched the Male Infertility Knowledgebase (http://mik.bicnirrh.res.in) for previous evidence on the relevance of the genes kept to fertility. We assessed the effect of each deleterious single-nucleotide variant (SNV) on protein functionality using the Ensembl Variant Effect Predictor (VEP) at https://www.ensembl.org/Tools/VEP, accessed on 3 November 2022. We additionally report effect predictions according to SIFT [87] and PolyPhen [88] as retrieved via Ensembl VEP. The databases mentioned before were accessed in 1 October 2022.

### 4.5. Network Analysis

Subsequently, we conducted network reconstruction to examine if the genes resulting from the above filtering steps might be part of a larger protein–protein interaction (PPI) network. For this purpose, we ran STRING v11.5 [89] at https://string-db.org/, using standard settings. The first reconstruction was confined to proteins for which variants associated with differential male fertility levels according to present GWAS. In a second approach, we additionally considered genes that were previously mentioned in the context of male fertility impairment. These were collected from the NCBI OMIM (Online Mendelian Inheritance in Man) database (state 1 February 2022). Corresponding OMIM entries related to male infertility, asthenozoospermia, teratozoospermia, and spermatogenetic failure. We focused on the largest connected component (LCC) resulting from the expanded sample of genes or proteins. Especially, we tested for PPI enrichment in the LCC and examined the number of PPIs per node (node degree). 

## 5. Conclusions

Based on the exomes of 100 men differing in fertility levels, we identified 12 associated SNPs localizing to nine genes (Figure 1). As shown, the nine genes focused are unlikely to have a direct impact on male fertility maintenance. They seem to acquire such relevance through the interaction with fertility-related genes or proteins instead. This results from present network reconstruction (Figure 4) and gains additional support from the effects predicted for the SNPs (Table 2). Thus, eleven of these SNPs imply amino acid exchanges, which should affect protein functionality moderately. Only one of these SNPs, rs200079869, would have larger consequences as it might interfere splicing of *HLA-DRB1* transcripts. However, whether splicing is really impaired remains to be examined, and, if so, it still may not have dramatic consequences for male fertility. In fact, theoretical considerations hardly suggest anything else than a weak effect for SNPs influencing male fertility. This is because alleles causing infertility cannot, by their very nature, be passed on to the next generation—at least not without medical assistance. Revisiting the results of previous GWASs seems to confirm this view. For example, Aston et al. (2010) [28] concluded from their GWAS that the SNPs which they found to be associated with oligozoospermia and/or azoospermia will unlikely play a significant role in spermatogenic failure. Likewise, Kosova et al. (2012) identified SNPs that associated with differential fecundity in men [31]. Yet, regardless of the quantitative differences, all men included in that GWAS had sired offspring prior to the study. Furthermore, in livestock, GWAS is usually performed against a background of differential, but nonetheless given, fertility [90,91]. Thus, the chances of identifying a central player in male infertility by GWAS could be limited in general. Nevertheless, GWAS-identified SNPs and the genes containing these can provide useful clues to male fertility markers. This might also be true for the SNPs and genes emerging from present GWAS, in subjects of central European origin at least. It may prove advantageous here that we had assembled a variety of disorders in the infertile cohort. Thus, the markers should have predictive potential on a wide range of disorders affecting sperm quality and quantity. This can now be tested in independent trials, e.g., by assessing transcript abundances in differentially fertile men by quantitative PCR. 

## Figures and Tables

**Figure 1 ijms-24-00524-f001:**
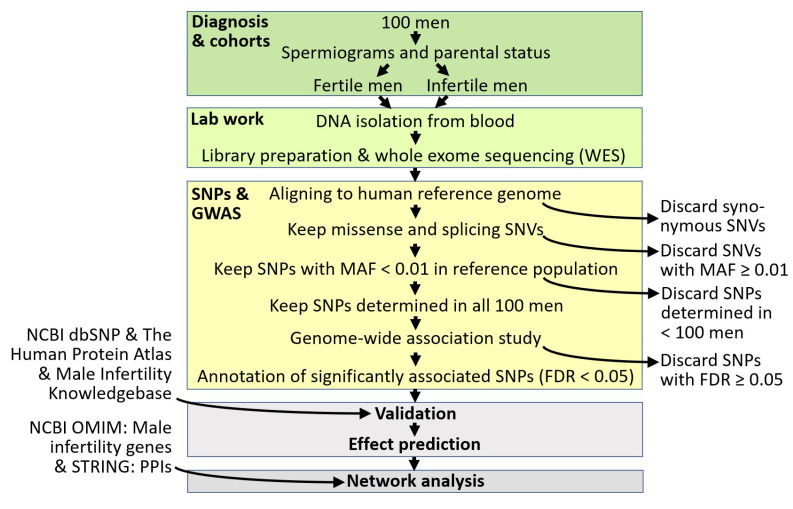
Workflow of the present study. Filtering steps leading to the discard of data are given on the right, and downstream inclusion of data from databases (NCBI dbSNP, The Human Protein Atlas, Male Infertility Knowledgebase, NCBI OMIM, STRING) on the left. Bold type relates to Section 4. For more details, see there. Abbreviations: FDR—false discovery rate; MAF—minor allele frequency; NCBI OMIM—Online Mendelian Inheritance in Man database at NCBI; NCBI dbSNP—SNP database at NCBI; PPI—protein–protein interaction; SNP—single-nucleotide polymorphism; SNV—single-nucleotide variant; GWAS—genome-wide association study.

**Figure 2 ijms-24-00524-f002:**
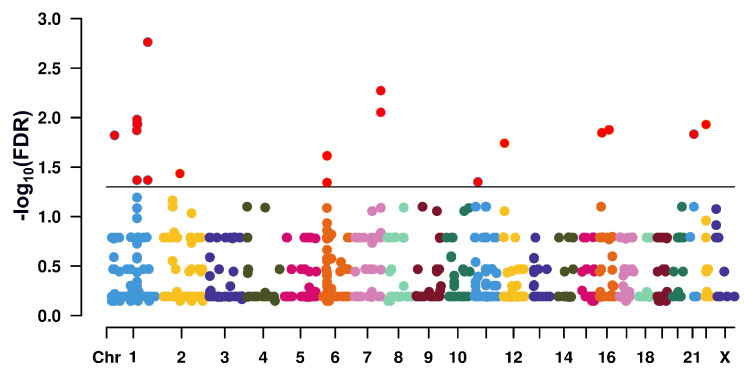
Manhattan plot giving the genomic position of single-nucleotide polymorphisms (SNPs) with differing allele frequencies between normal-fertile and infertile men. Only SNPs (N = 840) which had been genotyped in all 100 probands were considered. The Y-axis refers to —log10-transformed false discovery rates (FDRs). The horizontal line at 1.3 corresponds to the 5% significance threshold. SNPs exceeding the threshold are highlighted in red. Coloration of the remainder SNPs reflects their localization on different chromosomes. For clarity, chromosome (Chr) numbering is incomplete.

**Figure 3 ijms-24-00524-f003:**
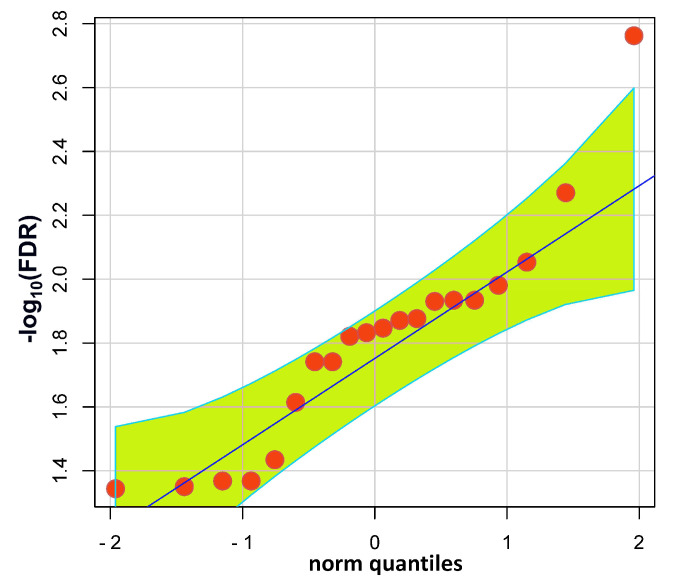
Q–Q plot of SNPs exceeding the 5% significance threshold of —log10-transformed FDR values (>1.3), suggestive of an association with fertility differences in 100 men. The blue diagonal gives the expectation under a normal distribution. The greenish area with turquoise boundaries corresponds to the 95% confidence interval.

**Figure 4 ijms-24-00524-f004:**
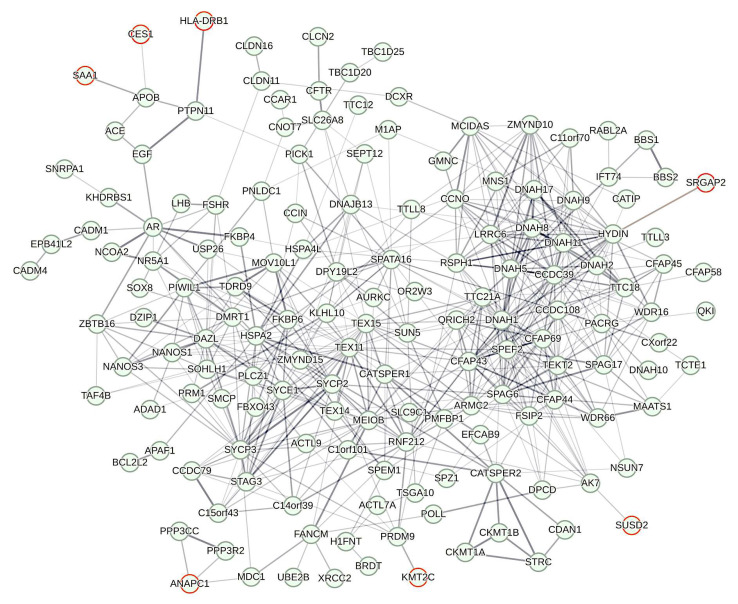
Largest connected component (LCC) of the protein–protein interaction (PPI) network reconstructed from genes implicated in male infertility. Most of the proteins included refer to genes reported in NCBI OMIM for their involvement in male infertility. Proteins to seven out of nine genes emerging from present GWAS (red circles) are included in the LCC, thereby occupying peripheral positions. Correspondingly, the number of PPIs per each of these genes is 1.286, whereas average node degree is 3.628 across the entire LCC. The LCC has 166 nodes between which 566 PPIs extend. Compared to the expected number of edges (N = 79) this represents a highly significant enrichment (*p* < 1 × 10^−16^). Average clustering coefficient across the LCC is 0.518. Strength of edges gives increments of minimum confidence (0.150, 0.400, 0.700, 0.900). Network reconstruction was conducted with STRING v11.5 using standard settings.

**Table 1 ijms-24-00524-t001:** Cohort characteristics inclusive of spermiogram parameters.

Parameter	Infertile Cohort *	Fertile Cohort *	Ratio
Age	35.5	31.5	1.1
Body-mass index	26.3	24.6	1.1
Semen pH	8.3	8.2	1.0
Spermatozoa (mio/mL)	5.3	60.7	0.1
Total spermatozoa amount (mio)	14.0	272.1	0.1
Motile spermatozoa (%)	27.0 **	50.0	0.5
Immotile spermatozoa (%)	55.0 **	34.0	1.6
Normal morphology (%)	4.0 **	12.5	0.3

The study included 70 infertile and 30 normal-fertile men. * Shown are median values. ** These sperm parameters were derived excluding five azoospermia cases. For more details, see Section 4.

**Table 2 ijms-24-00524-t002:** Single-nucleotide polymorphisms with different allele frequencies in infertile and fertile men.

				Infertile Probands			Fertile Probands					
Gene Name	SNP ID	REF	ALT	N_REF_	N_ALT_	MAF	N_REF_	N_ALT_	MAF	FDR	Function	Effect
ANAPC1	rs201128090	G	A	51	19	0.27	30	0	0.00	<0.05	missense	moderate
CES1	rs3826190	C	A	21	49	0.70	22	8	0.27	<0.05	missense	moderate
FAM131C	rs77667563	G	A	25	45	0.64	23	7	0.23	<0.05	missense	moderate
HLA-DRB1	rs3830125	C	T	44	26	0.37	29	1	0.03	<0.05	missense	moderate
HLA-DRB1	rs200079869	A	G	47	23	0.33	29	1	0.03	<0.05	splicing	high
KMT2C	rs2479172	C	T	16	54	0.77	21	9	0.30	<0.01	missense	moderate
KMT2C	rs183684706	A	G	36	34	0.49	28	2	0.07	<0.01	missense	moderate
NOMO1	rs62038492	A	G	24	46	0.66	23	7	0.23	<0.05	missense	moderate
SAA1	rs1136747	T	C	69	1	0.01	23	7	0.23	<0.05	missense	moderate
SRGAP2	rs782625719	G	C	24	46	0.66	26	4	0.13	<0.01	missense	moderate
SRGAP2	rs201036189	C	T	11	59	0.84	15	15	0.50	<0.05	missense	moderate
SUSD2	rs62231981	G	A	13	57	0.81	18	12	0.40	0.01	missense	moderate

Inference of false discovery rate (FDR) values followed the procedure by [43]. Abbreviations: ALT—alternative allele; MAF—minor allele frequency; N_ALT_—number of observations of the alternative allele; N_REF_—number of observations of the reference allele; REF—reference allele; SNP ID—single-nucleotide polymorphism identifier as verified in the dbSNP database of the National Center for Biotechnology Information (NCBI).

**Table 3 ijms-24-00524-t003:** Evidence suggesting fertility-relevance of genes harboring SNPs from present GWAS.

Gene	mRNA	Protein	Male Infertility Knowledgebase	Evidence	Reference
*ANAPC1*	T,E,P,SV	T,E,SV	aberrant CpGs in low motility sperm, cryptorchidism	Me, CpG	[44] [45]
*CES1*	T,E,P,SV	T	cryptorchidism, teratozoospermia	Me	[45]
*FAM131C*	T,E,P,SV		cryptorchidism, teratozoospermia	Me	[45]
*HLA-DRB1*	T,E,P,SV		azoospermia, cryptorchidism, gonadal dysgenesis, immunoinfertility, male infertility, sperm autoantibodies, spermatogenesis defects	Al	[46] [47]
*KMT2C*	T,E,P,SV		cryptorchidism, teratozoospermia	Me, Tr	[48] [45]
*NOMO1*	T,P,SV	T,E,P,SV	cryptorchidism	Me	[49] [45]
*SAA1*	T,E,P,SV		male subfertility, aberrant CpGs in low motility sperm	CpG	[44]
*SRGAP2*	T,E,P,SV	T,E,P,SV	spermatogenic defects, teratozoospermia	Tr	[50] [48]
*SUSD2*	T,E,P,SV	T,E,P,SV	cryptorchidism	Me	[45]

Expression data (mRNA, protein) refers to The Human Protein Atlas. Abbreviations: Al—allele frequencies; CpG—CpG pattern; E—epididymis; Me—methylation; P—prostate; T—Testis; Tr—transcript abundance; SNP—single-nucleotide polymorphism; SV—seminal vesicle.

## Data Availability

The datasets have been deposited at NCBI.

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
