# Peer review of "Genome-Wide Association Screening Determines Peripheral Players in Male Fertility Maintenance"

_ijms, 2022, doi:10.3390/ijms24010524_

Round 1

Reviewer 1 Report

The manuscript is very informative; however, the authors mentioned a few times in discussion that the genes that they identified possibly related to male fertility indirectly. The authors may try to strengthen their findings with more studies in depth and with literature supports.

Author Response

The manuscript is very informative; however, the authors mentioned a few times in discussion that the genes that they identified possibly related to male fertility indirectly. The authors may try to strengthen their findings with more studies in depth and with literature supports.

> Response: We thank Reviewer 1 for the suggestion above, which we have gladly considered. Thus, we have rewritten large parts of the manuscript including the Introduction, Results, and Discussion, considering much additional literature. In this way, we believe to have made clearer than before that genes identified by genome-wide association study (GWAS) tend to play a marginal and thus indirect role, rather than a central and direct role, in the maintenance of male fertility. Indeed, we found this expectation articulated in the work of other authors (e.g., Aston et al. 2010: see manuscript for detailed data). We also bring additional evidence according to which the SNPs we identified reside in genes with an indirect impact on male fertility. To have illustrated this trend, we consider one of the main merits of our study. We have additionally incorporated a workflow scheme (Figure 1) and, to the benefit of visualization of our analyses, Manhattan and QQ plots (Figures 1-3). Not least, we incorporated additional data and analyses characterizing the proband cohorts. Further analyses were opposed to our efforts, if not to keep the original resubmission deadline of 10 days, then at least not to exceed it too much. Thank you.

Reviewer 2 Report

The study provides novel insights regarding about the usefulness of using the SNPs located in the 9 genes studied (ANAPC1, 18 CES1, FAM131C, HLA-DRB1, KMT2C, NOMO1, SAA1, SRGAP2, and SUSD2) as markers of male fertility. The manuscript is clearly written, tables are adequate, and references are relevant. The work is well presented and written in a logical way. The experimental design is well done, and the results support the main conclusions of the manuscript. I would like to make some suggestions to help improve the manuscript.

Abstract:

-          I suggest that the authors indicate the meaning of the acronym of ONIM

Introduction:

-         The introduction is too brief, I recommend that the authors add information about the importance of clarifying the origin of male infertility and the results obtained so far through genome-wide association screening in relation to sperm quality.

These articles may be helpful:

Aston, K.I. and Carrell, D.T. (2009), Genome-Wide Study of Single-Nucleotide Polymorphisms Associated With Azoospermia and Severe Oligozoospermia. Journal of Andrology, 30: 711-725. https://doi.org/10.2164/jandrol.109.007971

Genome-Wide Expression of Azoospermia Testes Demonstrates a Specific Profile and Implicates ART3 in Genetic Susceptibility
Okada H, Tajima A, Shichiri K, Tanaka A, Tanaka K, et al. (2008) Genome-Wide Expression of Azoospermia Testes Demonstrates a Specific Profile and Implicates ART3 in Genetic Susceptibility. PLOS Genetics 4(2): e26. https://doi.org/10.1371/journal.pgen.0040026

Genome-wide Association Study Identifies Candidate Genes for Male Fertility Traits in Humans  Kosova, Gülüm et al. The American Journal of Human Genetics, Volume 90, Issue 6, 950 - 961

Results:

-          There are quite a few acronyms written for the first time without describing (WES, MAF, SNPs, LCC...). It will surely be due to the movement of the Materials and Methods section at the end of the manuscript. Please review all acronyms throughout the text so that they are well described.

-        I strongly encourage the authors to indicate the means of the percentages of the seminogram parameters (volume, pH, morphology, concentration, motility, viability...) of the different groups (control, azoospermia, oligozoospermia, asthenozoospermia, oligoasthenoteratozoospermia) in a table at the beginning of the results section.

 -       In relation to infertile patients, have you found significant differences in relation to the number of observations of the alternative allele between patients with azoospermia, oligozoospermia, asthenozoospermia or oligoasthenoteratozoospermia? This information would also be interesting to include in the results.

Discussion:

-          In the case of including information about the prevalence of the alternative allele in the different conditions of infertile patients (azoospermia, oligozoospermia, asthenozoospermia or oligoasthenoteratozoospermia), discuss what could be the cause.

Materials and Methods:

-          You must add more information about the commercial companies of the reagents and the software of the programs used.

For instance: (Qiagen, Hilden, Germany)

Conclusions:

-          The last sentence is quite risky, (line 250-251: To the best of our 250 knowledge, no such usability has been suggested before.) since there are articles that suggest the usefulness of genome-wide association screening to identify the genetic causes of male infertility. I recommend removing it

Author Response

The study provides novel insights regarding about the usefulness of using the SNPs located in the 9 genes studied (ANAPC1, 18 CES1, FAM131C, HLA-DRB1, KMT2C, NOMO1, SAA1, SRGAP2, and SUSD2) as markers of male fertility. The manuscript is clearly written, tables are adequate, and references are relevant. The work is well presented and written in a logical way. The experimental design is well done, and the results support the main conclusions of the manuscript. I would like to make some suggestions to help improve the manuscript.

Abstract:

-          I suggest that the authors indicate the meaning of the acronym of ONIM

> Response: We thank Reviewer 2 for this and the other comments which we believe to have matched with the revised draft of our manuscript.

Introduction:

-         The introduction is too brief, I recommend that the authors add information about the importance of clarifying the origin of male infertility and the results obtained so far through genome-wide association screening in relation to sperm quality.

These articles may be helpful:

Aston, K.I. and Carrell, D.T. (2009), Genome-Wide Study of Single-Nucleotide Polymorphisms Associated With Azoospermia and Severe Oligozoospermia. Journal of Andrology, 30: 711-725. https://doi.org/10.2164/jandrol.109.007971

Genome-Wide Expression of Azoospermia Testes Demonstrates a Specific Profile and Implicates ART3 in Genetic Susceptibility

Okada H, Tajima A, Shichiri K, Tanaka A, Tanaka K, et al. (2008) Genome-Wide Expression of Azoospermia Testes Demonstrates a Specific Profile and Implicates ART3 in Genetic Susceptibility. PLOS Genetics 4(2): e26. https://doi.org/10.1371/journal.pgen.0040026

Genome-wide Association Study Identifies Candidate Genes for Male Fertility Traits in Humans Kosova, Gülüm et al. The American Journal of Human Genetics, Volume 90, Issue 6, 950 - 961 

> Response: We gladly accomplished the proposed revision of the Introduction section. It now includes the papers suggested by Reviewer 2. We have gone well beyond this and have incorporated numerous other publications. In fact, we have rewritten the entire Introduction and believe that the section, in its current form, gives a comprehensive overview of the GWAS evidence on the causes of male infertility and the importance of further elucidation of these causes. In this context, we would like to note that we have also added numerous publications to the Discussion, all of which emphasize an indirect influence of the genes we found on the maintenance of male fertility.

Results:

-          There are quite a few acronyms written for the first time without describing (WES, MAF, SNPs, LCC...). It will surely be due to the movement of the Materials and Methods section at the end of the manuscript. Please review all acronyms throughout the text so that they are well described.

> Response: In the revised draft, all acronyms should be explained in the place where they are mentioned for the first time. We hope that we have not overlooked anything.

-        I strongly encourage the authors to indicate the means of the percentages of the seminogram parameters (volume, pH, morphology, concentration, motility, viability...) of the different groups (control, azoospermia, oligozoospermia, asthenozoospermia, oligoasthenoteratozoospermia) in a table at the beginning of the results section.

> Response: We have included a corresponding summary table (Table 1 in revised manuscript draft).

-       In relation to infertile patients, have you found significant differences in relation to the number of observations of the alternative allele between patients with azoospermia, oligozoospermia, asthenozoospermia or oligoasthenoteratozoospermia? This information would also be interesting to include in the results.

> Response: In addition to comparing normozoospermia vs. all infertility diagnoses considered (azoospermia, oligozoospermia, asthenozoospermia, oligoasthenozoospermia, and oligoasthenoteratozoospermia), we had compared normozoospermia with a reduced infertility group lacking azoospermia cases. By its very nature, this procedure had no effect on the allele frequencies in the normozoospermia group. But the allele frequencies in the infertility group also showed only minor deviations in the second decimal place compared with the values reported in Table 2 for the complete infertility cohort. Furthermore, the infertility diagnoses in the smaller infertility group (the one lacking azoospermia cases) continued to cover the factors sperm count, sperm motility, and sperm shape. This led us to abstain from inclusion of the second comparison in the revised manuscript.

Discussion:

-          In the case of including information about the prevalence of the alternative allele in the different conditions of infertile patients (azoospermia, oligozoospermia, asthenozoospermia or oligoasthenoteratozoospermia), discuss what could be the cause.

> Response: Thanks for the suggestion but see above, please.

Materials and Methods:

-          You must add more information about the commercial companies of the reagents and the software of the programs used.

For instance: (Qiagen, Hilden, Germany)

> Response: In the revised draft, we provide more information on companies and software.

Conclusions:

-          The last sentence is quite risky, (line 250-251: To the best of our 250 knowledge, no such usability has been suggested before.) since there are articles that suggest the usefulness of genome-wide association screening to identify the genetic causes of male infertility. I recommend removing it

> Response: The comment made it clear to us that the sentence in question was misleadingly worded. Our intention was not to point out as an innovation the usability of GWAS for elucidating the molecular background of male infertility. Rather, we meant to propose the SNPs and genes resulting from present GWAS as male fertility markers for the first time. Anyway, we have followed the suggestion of Reviewer 2 and deleted the sentence. Instead, we write in the revised draft that the potential of the SNPs and the corresponding genes as male fertility markers can now be validated in independent approaches. Thanks a lot. 

Reviewer 3 Report

The authors have analysed the  WGAS on the exomes of 100 men and looked  for new gene targets associated with infertility conditions.

The study i well designed properly conducted. However ,  still some improvement are needed to be published in the journal.

 For methods and materials better to include a flow diagram  for the strategy used with all criterion followed.  

Under the results as a supplement , please provide the data on study cohort charateristi likes age, BMI, any clinical triats/sperm /ejaculate characteristics in a summary  table format .

Please  show the Annotation of Validation SNPs and Association Statistics with  sperm or other parameters used with the cohort . Proabaly can use Manhattan and Q-Q plots

Have the authors validated the genes found using  validation cohort using qPCR or any other  means? if so please provide the data , ( other than in silico validation)

The authors may have missed a nice study  performed on a Japanese cohort from Sato et al ., , please discuss the present  outcomes with those.  

Author Response

The authors have analysed the  WGAS on the exomes of 100 men and looked  for new gene targets associated with infertility conditions.

The study i well designed properly conducted. However ,  still some improvement are needed to be published in the journal.

 For methods and materials better to include a flow diagram  for the strategy used with all criterion followed.

> Response: We are grateful for these and the other suggestions. New Figure 1 is a flow diagram giving the strategy applied and the criteria which resulted in the SNPs and genes reported in the manuscript.

Under the results as a supplement , please provide the data on study cohort charateristi likes age, BMI, any clinical triats/sperm /ejaculate characteristics in a summary  table format .

> Response: The revised manuscript contains a new Table (Table 1) which yields the respective cohort characteristics.

Please  show the Annotation of Validation SNPs and Association Statistics with  sperm or other parameters used with the cohort . Proabaly can use Manhattan and Q-Q plots

> Response: Inspired by this comment, the revised manuscript contains two new Figures. The Manhattan plot in new Figure 2 yields an overview of the genomic distribution of the SNPs determined and the corresponding false discovery rates (FDRs). New Figure 3 is a Q-Q plot demonstrating that the FDRs of almost all significantly associated SNPs are within the 95% confidence interval of the expected normal distribution, indicative of the independence of the corresponding test results. As described in more detail in the revised "Results" and "Materials and Methods" sections, we subjected these SNPs to manual validation. Using the dbSNP database at NCBI, we verified the single-nucleotide variants (SNVs) involved, genomic position (chromosomes, nucleotide positions), and names of the corresponding genes. To be kept, the genes additionally had to be expressed in male reproductive tissues according to respective data at The Human Protein Atlas. These filtering steps are additionally given in new Figure 1. Further data on annotation and test results are reported in Supplementary Table S2. With these additions and explanations, we hope to have properly matched your comment.

Have the authors validated the genes found using  validation cohort using qPCR or any other  means? if so please provide the data , ( other than in silico validation)

> Response: We share the view that the SNPs and genes identified can now be further tested as biomarkers of male fertility and infertility. However, the original resubmission deadline of 10 days seemed to preclude the performance of new analyses. (Even if we were not able to meet the said submission deadline due to teaching obligations starting with the winter semester, we at least wanted to exceed it as little as possible). Still, the comment raised above led us rephrase the end of the Conclusions. In the revised draft, we finish with stating that the suitability of the SNPs and genes resulting from present GWAS as male fertility markers can now be validated e.g., by qPCR. The second achievement of our study besides the advancement of diagnostics, namely, to have shown that genes identified by GWAS are increasingly of marginal importance for the maintenance of male fertility, remains in our opinion unaffected by this reservation.

The authors may have missed a nice study  performed on a Japanese cohort from Sato et al ., , please discuss the present  outcomes with those.  

> Response: This and other studies have been included into the study. Thank you.

Round 2

Reviewer 2 Report

The authors have addressed the concerns and incorporated them into the manuscript.